The challenge of evaluating pain and a pre-incisional local anesthetic block

McKune Carolyn M. 1 4 mckune@mythosvet.com
Pascoe Peter J. 1
Lascelles B. Duncan X. 2
Kass Philip H. 3
1 Department of Surgical and Radiological Sciences, School of Veterinary Medicine, University of California , Davis, CA , USA
2 Comparative Pain Research Laboratory, Department of Clinical Sciences & Center for Comparative Medicine and Translational Research, College of Veterinary Medicine, North Carolina State University , Raleigh, NC , USA
3 Department of Population Health and Reproduction, School of Veterinary Medicine, University of California , Davis, CA , USA
Jones Philip
4 Current affiliation: Mythos Veterinary LLC, Gainesville, FL, USA

Electronic publication date: 2014 Apr 10
Publication date: 2014
Volume: 2
Electronic Location ID: e341
Received 2013 Dec 31; Accepted 2014 Mar 23
Copyright: © 2014 McKune et al.
Copyright year: 2014
Copyright holder: McKune et al.
License: This is an open access article distributed under the terms of the Creative Commons Attribution License, which permits unrestricted use, distribution, and reproduction in any medium, provided the original author and source are credited.
License URL: https://creativecommons.org/licenses/by/3.0/

Keywords: Bupivacaine, Lidocaine, Pain assessment, Local anesthesia, Dog

Funding: Center for Companion Animal Health, University of California (Davis) This work was funded by the Center for Companion Animal Health, University of California (Davis). The funders had no role in study design, data collection and analysis, decision to publish, or preparation of the manuscript.

==============================
Background. Our objective was to test the effectiveness of a local anesthetic line block administered before surgery in reducing postoperative pain scores in dogs undergoing ovariohysterectomy (OVHX).

Methods. This study is a prospective, randomized, blinded, clinical trial involving 59 healthy female dogs. An algometric pressure-measuring device was used to determine nociceptive threshold, and compared to three subjective pain scales. Group L/B received a line block of lidocaine (4 mg/kg) and bupivacaine (1 mg/kg) subcutaneously in the area of the incision site and saline subcutaneously as premedication; group L/BM (positive control) received a similar block and morphine (0.5 mg/kg) subcutaneously for premedication; and group SS (negative control) received a saline line block and saline premedication. Criteria for rescue analgesia were defined before the study. Dogs were assessed prior to surgery, at extubation (time 0) and at 2, 4, 6, 8 and 24 h post-recovery. The data were analyzed with one-way ANOVA, and a Split Plot Repeated Measures ANOVA with one grouping factor and one repeat factor (time). P < 0.05 was considered statistically significant.

Results. Approximately 33% of dogs required rescue analgesia at some point during the study, with no significant difference between groups. There was no significant difference between treatment groups with any assessment method.

Conclusions. As there were no statistically significant differences between positive and negative controls, the outcome of this technique cannot be proven.

Introduction

As any verbal responder who has experienced pain may attest to, pain decreases quality of life (Niv & Kreitler, 2001). Therefore, pain management in patients experiencing pain is crucial for improving quality of life. Pain management of non-verbal patients is uniquely challenging because the ability to effectively diagnose and treat pain becomes very subjective. Pain assessment in non-verbal species has been investigated along three principal lines: (a) objective measures of physiologic responses to experimental pain, (b) subjective or semi-objective assessment of behavior postoperatively, and (c) quantitative measures of postoperative behavior and physiology. While studies using objective physiological data (i.e., variables such as heart rate, respiratory rate and blood pressure) are easy to perform and analyze statistically, there is minimal evidence that these measures are reliable indicators of pain (Cambridge et al., 2000; Conzemius et al., 1994). Most peer-reviewed research studies in veterinary medicine use subjective or semi-objective assessments of postoperative pain or sensitivity of an anatomical site to assess outcomes.

Algometers are devices used to quantitate pressure required to elicit a response from a subject; this is termed “nociceptive threshold”. Algometers provide a (partially) objective measurement of incisional sensitivity. The “threshold” reading is numeric and objective, but the factor determining the threshold (behavioral response) is subjective. Various mechanical threshold devices are validated to assess somatosensory processing changes (Briley et al., 2014).

Multimodal analgesia is the combination of analgesic drugs with different methods of action, with the goal of reducing or preventing nociceptive stimulation at multiple receptors and pathways. In humans, multimodal analgesia has been shown to decrease post-operative morbidity and mortality, improve quality of life and patient satisfaction, and decrease the associated costs to hospitals and insurance companies (Skinner, 2004). In addition to the general agreement of a clinical benefit to this approach (Lascelles & Main, 2002), there are also an increasing number of research studies in non-verbal species supporting multimodal analgesia (Lascelles et al., 2008; Slingsby, Murrell & Taylor, 2010; Martins et al., 2010). One simple way to include multimodal analgesia is the incorporation of a local anesthetic to desensitize a specific region, in combination with systemic analgesic administration.

This study was designed to assess the effect of pre-incisional administration of a combination of local anesthetics on post-operative pain, measured by subjective and objective pain scores after canine ovariohysterectomy (OVHX). We hypothesized that pre-incisional infiltration of the incision area with local anesthetic agents (group L/B) would result in similar post-surgical pain levels compared to animals receiving local anesthetic and an opioid (group L/BM), and decreased post-surgical pain compared to animals not receiving any pre-operative analgesics (group SS).

Materials and Methods

This study examined 59 healthy intact female dogs admitted to a local animal shelter (Sacramento Society for Prevention of Cruelty to Animals [SPCA], Sacramento, CA, USA), ranging in age from six months to eight years old with weights ranging from 3.4 to 35.5 kg. A physical examination was performed, and temperature, heart rate, and respiratory rate were recorded prior to sedation for anesthesia and surgery. Each dog had a packed cell volume (PCV), total protein (TP), and blood urea nitrogen (Azostick; Bayer Corporation, Elkhart, IN, USA) checked prior to surgery. Please see Table 1 for a summary of baseline data. No dogs with abnormal physiologic parameters, abnormal blood tests, evidence of a previous OVHX, or requiring extension of the incision beyond the blocked area were used in this study. All protocols were approved by the University of California, Davis, Institutional Animal Care and Use Committee, as well as by administrative study reviewers at the Sacramento Society for Prevention of Cruelty to Animals (SSPCA).

Table 1 Baseline data for Groups L/B, L/BM, and SS. Data is presented as average (±SD), except for BUN, where average value only is listed.

Respiratory rate was not included because a large number of animals were panting.

Group	L/B	L/BM	SS	
Number of dogs	20	19	20	
Age (years)	1.6 ± 1.7	1.6 ± 1.4	2.3 ± 2.0	
Weight (kg)	17 ± 6.8	16.5 ± 1.4	18.2 ± 9.6	
Temperature (F)	101.2 ± 1.0	101.1 ± 0.9	101.1 ± 1.0	
Heart rate (BPM)	140 ± 22	138 ± 26	138 ± 22	
PCV (%)	43 ± 4.0	42 ± 4	42 ± 4	
Total protein (g/dL)	6.8 ± 0.6	6.8 ± 0.7	6.5 ± 0.6	
BUN (Azostick)	5–15	5–15	5–15	
Propofol (mg/kg)	4.6 ± 1.1	4.3 ± 1.6	3.6 ± 1.6	

Anesthesia

Dogs were allocated into one of three groups using a computer generated randomized block design. All three groups were sedated with acepromazine (Acepromazine maleate, Vedro, St. Joseph, MO, USA) (0.03 mg/kg, subcutaneously [SC]) administered prior to catheter placement. An 18–22-gauge (depending on the animal’s weight) over the needle IV catheter was placed in a cephalic vein for drug and fluid administration. Anesthesia was induced with propofol (Diprivan; AstraZeneca LP, Wilmington, DE, USA) to effect and maintained with isoflurane (Isoflurane; Abbot Laboratories, North Chicago, IL, USA) in oxygen to effect. Lactated Ringer’s solution was administered at 10 ml/kg/h until recovery. Heart rate, respiratory rate, and systolic blood pressure were monitored throughout the procedure.

Dogs in group L/B received a line block prior to surgery in the incision area, consisting of 4 mg/kg lidocaine (Lidocaine; Hospira Inc., Lake Forest, IL, USA) and 1.0 mg/kg bupivacaine (Bupivacaine; Hospira Inc., Lake Forest, IL, USA). These dogs also received 0.05 mg/kg of saline SC at the same time as acepromazine administration. Group L/B were test subject dogs, to compare to positive and negative control groups. Dogs in the group L/BM received a line block prior to surgery, using 4.0 mg/kg lidocaine and 1.0 mg/kg bupivacaine. These dogs also received 0.5 mg/kg of morphine (Morphine sulfate; Baxter Health Care Corporation, Deerfield, IL, USA) SC at the same time as acepromazine administration. Group L/BM was the positive control group (i.e., dogs anticipated to have minimal pain). Group SS was the negative control group (i.e., dog anticipated to have pain). Dogs in group SS received 0.275 ml/kg of normal saline prior to surgery in the incisional area. These dogs also received 0.05 mg/kg of saline SC at the same time as acepromazine administration. Because we anticipated animals with pain, criteria for rescue analgesia were defined prior to the study’s commencement and strictly adhered to. The line block or saline (depending on the group) was administered after induction of anesthesia and initial surgical preparation of the field, approximately five minutes prior to surgical incision.

Line block procedure

Appendix 1 shows the line block in schematic form. Local anesthetic or saline (depending on the group) was infused with a 2.5 inch, 22-gauge spinal needle in three separate lines to form an inverted double “L” administration site. One third of the volume of drug or saline was administered at each site, as volume allowed. The level of the first line (Appendix 1, “1”) was roughly halfway between the umbilicus and the first set of nipples below the umbilicus; placement was guided by consultation with the surgeon prior to incision to ensure coverage of the area to be incised (Appendix 1, “A”). The width of this first line ran mediolaterally for approximately 1.25 cm on either side of midline. The second line (Appendix 1, “2”) began at the left-most lateral point of the first line, and ran craniocaudally for the length of the spinal needle on the left side of midline. The third line (Appendix 1, “3”) paralleled the second on the right side of the umbilicus. In Appendix 1, “B” denotes the pubis. These blocks were administered in the subcutaneous and fascial planes. Aspiration prior to administration of the block was performed to ensure the drugs were not given intravenously.

Surgical procedure

The hair was clipped from the xiphoid process to the pubis and three cm laterally to the nipple on both sides of the abdomen. The skin was scrubbed with chlorhexedine and rinsed with water three times. The line block was applied after initial preparation; additional preparation followed until the area was aseptically prepared. An incision was made extending below the umbilicus to one-third the distance from the umbilicus to the pubis. An OVHX was performed in a standard fashion (Fossum, 2007) by one of three experienced, shelter veterinary surgeons. The skin was closed in a routine manner.

Assessment

Four pain scoring assessments were used; initial values for each were recorded prior to the sedation of the animal for anesthesia and surgery (time negative one). Assessments were then made at zero (time of extubation), two, four, six, eight, and 24 h postoperatively by one veterinarian (CMM) who was blinded to which treatment group each animal was in. Caretakers made additional assessments during the day when animals were handled, to ensure any animal that needed additional analgesia would receive it.

The first pain scoring assessment was a visual analog scale (VAS) score. This assessment was made prior to any manipulation or handling of the animal. A mark on a ten centimeter (cm) line corresponded to the assessor’s visual assessment of the animal’s pain, ranging from zero (“no pain”) to ten cm (“the most pain an animal could possibly be in”), measured in mm using a standard ruler at each scoring assessment, and recorded after each measurement was taken.

The next two pain scoring assessments were done sequentially. One of these pain scales was based on a previously validated scoring system, the Glasgow Composite Pain Scale (GCPS, http://www.gla.ac.uk/faculties/vet/smallanimalhospital/ourservices/painmanagementandacupuncture, subheading: Short form pain questionnaire). The primary variables included vocalization (quiet, crying, groaning, screaming), attention to painful area (ignoring, looking, licking, rubbing, or chewing), mobility (normal, lame, slow or reluctant, stiff, or refusal to move), response to touch (none, looking around, flinch, growl, snap, or cry), demeanor (happy and content, bouncy, quiet, non-responsive or indifferent to surroundings, nervous or anxious or fearful, or depressed or non-responsive to stimulation), and posture (comfortable, unsettled, restless, hunched or tense, or rigid). Additional assessment was made using the University of Melbourne Pain Scale (UMPS) (Firth & Haldane, 1999). The primary variables included physiologic data (dilated pupils, percentage increase in heart rate, percentage increase in respiratory rate, rectal temperature, salivation), response to palpation (no change, guards/reacts when touched, guards/reacts before touched), activity (at rest [sleeping or semiconscious, awake], eating, restless [pacing, getting up and down], or rolling/thrashing), mental status (submissive, overtly friendly, wary, or aggressive), posture (guarding or protecting affected area, recumbency, standing or sitting with head up, standing with head down, moving, or abnormal body posture [prayer/hunched]), and vocalization (none, vocalizing when touched, intermittent vocalization, or continuous vocalization).

The final assessment method used a digital von Frey apparatus (IITC 2390 Series Electronic Von Frey Anesthesiometer; Woodland Hills, CA, USA) (KuKanich, Lascelles & Papich, 2005a). The tip of the von Frey apparatus was placed one cm adjacent to the center of the incision. It was pressed with a slow, continuous pressure until a response was noted, with a maximal force of 1000 g. A response was considered an acknowledgement that the stimulus was noxious; this included behaviors such as withdrawing from the stimulus, a cry, active head turn to the stimulus, attempt to bite, etc. This measurement was repeated three times at five-minute intervals, and each value was recorded as force in grams. The average value of these three readings was used in the data analysis. At each time point, algometer measurements were also taken from the lateral thoracic wall in the same manner. These measurements, as well as pre-sedation measurements, acted as controls for analysis.

Rescue analgesia protocol

All animals were assessed by the observing veterinarian (CMM), and rescue analgesia (0.5 mg/kg morphine IM) was administered to any animal that achieved a maximum score in any one category of the GCPS, any animal with a pain score of 8 or greater on the GCPS or who did not improve over time as compared to pre-sedation GCPS score, any animal developing aggression, or a combination of these previous factors. Animal handlers at the SPCA also had the opportunity to declare an animal as being in pain, based on their observation, and these animals also received rescue analgesia. Administration of rescue analgesia and the reason for administration was recorded, and these animals were included in assessments; see “Blinding, exclusion criteria and statistical analysis”. Any animal receiving rescue analgesia was reassessed 30 min later to ensure efficacy of the rescue analgesia administration.

Blinding, exclusion criteria, and statistical analysis

The evaluator (CMM) was blinded to which dog was in which group (i.e., L/B, L/BM or SS) as well as to whether a placebo or a study drug was contained in a particular group. The statistician who performed the data analysis remained blinded to which study drug was contained in each group until the analyses were completed.

Initial power calculations were performed prior to commencing the study. An alpha error level was set at 5%. Standard deviation was set at 1.8 Glasgow Composite Pain Scale units (Morton et al., 2005). A beta error level was set at 20%. These calculations indicated the need for approximately 19 dogs in each group to find significant differences in our study populations, assuming a difference of 2.6 on the Glasgow Composite Pain Scale as being significant (Morton et al., 2005). The groups were analyzed for differences in age, weight, preoperative temperature, heart rate, respiratory rate, BUN, PCV/TS, propofol dose [mg/kg], and time negative one algometric values, by means of one-way ANOVA. Normality of the errors was assessed by visual inspection of a histogram of the errors and a normal probability plot. Errors were considered normal if the histogram was unimodal and approximately symmetrical (Petrie & Watson, 2006), and the normal probability plot was an upwardly sloping, approximately straight line. Homogeneity of variance was tested by means of a studentized residual vs. means plot. The response variable of treatment groups was analyzed by means of a repeated measures ANOVA with one grouping factor and one repeat factor (time). Those dogs receiving rescue analgesia were analyzed in a similar fashion in two separate analyses: within their collective treatment group and as a separate subgroup. P < 0.05 was considered statistically significant.

Results

There were 20, 19 and 20 dogs in Groups L/B, L/BM, and SS, respectively, for a total of 59 dogs. Twenty of the 59 dogs initially enrolled, required rescue analgesia (seven, three and ten dogs in groups L/B, L/BM, and SS, respectively, with no significant differences in the proportion requiring rescue analgesia between groups). Of all the predetermined rescue analgesia criteria, the only criteria triggering administration of rescue analgesia were animals that achieved a maximum score in any one category (mobility: refusal to move) of the GCPS and animals developing aggression. The majority of the dogs requiring rescue analgesia required it at time 0 (extubation; 18 of 20 dogs) for refusal to move. All fifty-nine dogs were included in the analysis; additional analysis of the separate subgroup of dogs who received rescue analgesia showed similar results to the analysis of all 59 dogs, but the low numbers of dogs remaining in the groups after removal of those requiring rescue analgesia brought into question the validity and precision of the statistical analyses (therefore, data not shown).

VAS, GCPS, and UMPS analyses showed no significant difference in pain scores between treatment groups, and there was a significant effect of time (i.e., a decrease in pain scores over time; Figs. 1, 2 and 3). Algometric values were compared to one of two controls. Regardless of whether the value obtained at the wound was compared to the thoracic measurement obtained at the same time or compared to the pre-incisional control reading (i.e., measurement at abdomen/control measure), there was no significant difference in values obtained between treatment groups, and there was a significant effect of time (i.e., a decrease in pain scores over time; Figs. 4 and 5).

Figure 1 Visual Analogue Scale (VAS), from 0 to 10 cm, prior to premedication (time −1), extubation (time 0), and 2, 4, 6, 8 and 24 h post-operatively.

Note: L/B received saline premedication and local anesthetic line block, L/BM received morphine premedication and a local anesthetic line block, and SS received a saline premedication and saline line block. Error bars represent standard deviation.

Figure 2 Glasgow composite pain scale (GCPS) scores from 0 to 24 prior to premedication (time −1), at extubation (time 0), and 2, 4, 6, 8 and 24 h post-operatively.

Note: L/B received saline premedication and local anesthetic line block, L/BM received morphine premedication and a local anesthetic line block, and SS received a saline premedication and saline line block. Error bars represent standard deviation.

Figure 3 University of Melbourne Pain Scale scores from 0 to 27 prior to premedication (time −1), at extubation (time 0), and 2, 4, 6, 8 and 24 h post-operatively.

Note: L/B received saline premedication and local anesthetic line block, L/BM received morphine premedication and a local anesthetic line block, and SS received a saline premedication and saline line block. Error bars represent standard deviation.

Figure 4 Algometric value, depicted as a ratio compared to the value obtained at the abdomen versus the value obtained at the thorax at the same time points: at premedication (time −1), at extubation (time 0), and at 2, 4, 6, 8 and 24 h post-operatively.

Note: L/B received saline premedication and local anesthetic line block, L/BM received morphine premedication and a local anesthetic line block, and SS received a saline premedication and saline line block. Also note that a ratio of one indicates the animal tolerates the same level of pressure on the abdomen as the thorax. A decreasing ratio indicates the animal tolerates less pressure on the abdomen as compared to the thorax.

Figure 5 Algometric value, depicted as a ratio comparing the value obtained at each individual time point to values obtained at the abdomen prior to premedication (i.e., time, but not location, is the dependent variable).

Time points for comparison to pre-medication values include pre-medication (time −1), extubation (time 0), and 2, 4, 6, 8 and 24 h post-operatively. Error bars represent standard deviation. Note: L/B received saline premedication and local anesthetic line block, L/BM received morphine premedication and a local anesthetic line block, and SS received a saline premedication and saline line block. Also note that a ratio of one indicates the animal tolerates the same level of pressure on the abdomen at the time of comparison as it tolerated prior to incision. A decreasing ratio indicates the animal tolerates less pressure on the abdomen at the time of comparison as compared to pressure applied prior to the incision.

Discussion

We chose three different groups to test the efficacy of our line block to improve postoperative pain scores and algometric values. One group of animals (L/BM) was selected to receive morphine premedication to serve as the positive control group (i.e., the group anticipated to have the best analgesia). The group of animals that did not receive analgesia (SS) served as the negative control (i.e., the group anticipated as having pain). The treatment group of interest, L/B, was evaluated in comparison to these positive and negative controls. The most profound result of our study was the lack of statistically significant differences between our positive and negative control at any given time point; that is, there was no statistically significant difference between an animal that received no preemptive analgesia and an animal receiving a full mu opioid receptor agonist to provide analgesia, using any of the assessment methods. This result was surprising, not only from the perspective of rendering the effects of treatment only speculative, but also in the implications this possesses for investigators researching pain in non-verbal species.

There are a number of potential reasons for the results obtained. Study design is critical to successfully identifying targeted outcome. One potential reason no significant difference between pain scores for any treatment group was evident was the number of dogs included in the study, thus limiting statistical power of our study. Our initial sample size calculations potentially hindered the study in two ways. Firstly, we applied sample size calculations meant for two groups to three groups. In retrospect, in order to correctly calculate our initial sample size, we would modify alpha (P = 0.05), with three groups and the number of potential comparisons (Conzemius et al., 1994), and therefore use an alpha value of 0.017 (0.05/3); this was not done. Secondly, our initial sample size calculations used a difference in the GCPS of 2.6, based on previous work (Morton et al., 2005). This was regarded as the minimum difference that would be clinically relevant. The differences in pain scores in our study were smaller than this (Fig. 2) and while increasing the number of animals treated may possibly have reached statistical significance it would still have had little relevance for the clinician. Additionally, because we cannot account for Type II error, our statistical analysis is not conclusive.

The other aspect of study design was the intent to maximize the potential for successful pain identification, and thus the inclusion of one group that did not receive any preemptive analgesic medication (negative control). This decision was not made lightly, and the criteria were very strict for the use of rescue analgesia because of this. Even in light of this group that intentionally included, albeit aggressively managed for, pain, there was still no significant difference between the negative and positive control groups.

It may be that the dogs in this study were experiencing little discomfort, making it difficult to distinguish between the treatment groups. While this may seem unreasonable in regards to an intra-abdominal procedure, pain scores on the only validated scoring system (GCPS) were very low, never achieving a score of greater than five out of a maximal value of 24 at any one time point. A study evaluating intervention levels using the GCPS suggested intervening if a score of greater than seven out of 24 was obtained; the GCPSs values obtained in the present study were below this threshold (Reid et al., 2007). With such low pain scores, it was difficult to establish differences between the treatment groups. The low pain scores may have been due to the highly experienced veterinarians who were performing the OVHX creating minimal tissue trauma during surgery (and thus minimal pain associated with the surgery). In this study, the three surgeons were shelter veterinarians who performed up to 40 surgeries on any given day with over 30 years of combined experience between them; surgery time ranged from 11 to 47 min, with an average surgery time of 21 min. This is considerably less than the average time of 140 min for a veterinary student to spay a dog (Kennedy, Tamburello & Hardie, 2011). If a group of less experienced surgeons—for example, veterinary student surgeons—performed the procedures, more detectable differences may have arisen. There is extensive debate about this subject, further complicated by a lack of reporting surgeon experience level in well-performed pain studies. At least one study specifically examining surgeon experience level suggested experience level of the surgeon was not correlated with a change in postoperative pain score (Wagner et al., 2008). However, recent basic science evidence underscores the importance of deep tissue trauma to the experience of pain (Xu & Brennan, 2010). Basic science work also supports this on a receptor level: surgical tissue injuries enhanced the membrane translocation of receptors important in post-operative hypersensitivity (Wang et al., 2013; Michelsen et al., 2012). Surgery performed by experienced surgeons, as was the case in this study, may reduce post-operative pain (Devitt, Cox & Hailey, 2005; Freeman et al., 2010) to levels below the sensitivity of current pain assessment scales.

Another reason for low pain scores on various scales may be due to inherent insensitivity of the measurement techniques, preventing a significant difference between positive and negative controls. Surprisingly little work has been performed to produce validated assessment systems for acute pain, with the Glasgow Composite Pain Scale standing out as the most validated scale in this regard (Morton et al., 2005). However, this scoring system was validated using a variety of surgical procedures, including orthopedic procedures. Additionally, the GCPS has not undergone criterion validation testing. It is possible that a dog undergoing OVHX by an experienced veterinarian may have signs of pain more subtle than this assessment instrument can detect. The von Frey apparatus was sensitive to changes in threshold testing with dogs given 1 mg/kg morphine (KuKanich, Lascelles & Papich, 2005a), and appears reliable in clinically normal dogs (Briley et al., 2014). However, data gathered by one of the authors (BDXL) found no difference in von Frey thresholds when it was used to assess wounds being infused with saline or with local anesthetic (Hardie et al., 2011). This suggests that the von Frey may not be the appropriate instrument for assessing sensitivity of clinical wounds. Testing site could make a difference in the reliability of the algometer, as previous reports suggest that the canine carpal pad may be the most satisfactory site for testing (KuKanich, Lascelles & Papich, 2005a; KuKanich, Lascelles & Papich, 2005b). Because this location was considered unusual for testing sensitivity of an abdominal wound, it was not used for either the control or the test site, which may contribute to the difficulty of using the algometer for assessment. This topic needs further research to understand why the results appear counterintuitive, and to understand appropriate means to assess wound sensitivity.

There is no doubt that expertise of the assessor in regards to pain assessment plays a major role, as evidenced by a single experienced anesthesiologist finding a statistically significant improvement after an incisional block with bupivacaine in dogs undergoing a celiotomy (Savvas et al., 2008). As involved as veterinarians are in the care of animals on a daily basis, it is still possible to misclassify an animal as not in pain for many reasons—including temperament, breed, type of surgery, and surgeon experience. In a study comparing staff observations versus a self-report of pain in young children, staff observations of pain were generally lower than the self-reports (Shavit et al., 2008). However, for animals there is little alternative to an observer for pain assessment. The negative aspects of such a misclassification are obvious. The inclusion of multiple pain assessment tools with very defined criteria was intended to counter potential inexperience, but cannot negate the possibility altogether. Although the differences in the three reduced-size groups that received rescue analgesia failed to reach statistical significance, the difference between the L/BM group and the SS group (16% vs. 50% treated), if real, is clinically important and suggests that the clinical judgment of when to administer rescue analgesia includes factors that are not captured in the scoring systems that were used. We elected to give rescue analgesia to any patient with a maximum value in any one GCPS category (Ahn et al., 2013; Tsai et al., 2013; Odette & Smith, 2013), as a means to favor generous administration of rescue analgesia for any patient who might need it. Our decision to give rescue analgesia to patients with a maximum value in any one GCPS category may have biased our results, as 18 of 20 dogs received rescue analgesia for a maximum value in the category of refusing to move post-surgery. However, given the large number of patients in group SS that received rescue analgesia (almost half of the animals in that group), it is possible that refusal to move may be a sensitive indicator of patient discomfort in the patient with pain secondary to an OVHX.

The effect of time present (i.e., a decrease in pain scores over time) in this study suggests that we do see changes in pain scale scores and von Frey readings over the course of a 24-h period. Using subjective pain scores, all values returned to baseline or near baseline by 24 h, suggesting that we could no longer detect pain effectively at that point. When assessing algometric scores, there was an initial decrease from baseline after extubation, and while values tended to move back towards baseline between eight and 24 h, the values never returned to baseline. This suggests wound sensitivity may still be present when subjective assessments do not detect pain. An alternative explanation is that the dogs had become behaviorally sensitized to the testing device. Ideally, testing of dogs that were not operated on would have been performed to evaluate the effect of time on threshold readings. Data (Coleman et al., 2011) suggest there is a learned response that decreases thresholds over time in normal dogs, but the data were generated using a more blunt device than the von Frey used in the present study.

No adverse events were documented in this study to suggest that a local anesthetic infiltrative block is harmful to a patient, as opposed to another study examining incisional line block (Fitzpatrick, Weir & Monnet, 2010). Fitzpatrick, Weir & Monnet (2010) may have seen greater complications because they choose to infiltrate the site of the incision, where as we infiltrated the tissue surrounding the incision. The block took a short time (<2 min) to perform. Other studies have found that incisional blocks provide effective analgesia (Savvas et al., 2008; Carpenter, Wilson & Evans, 2004).

Conclusions

We believe we cannot make firm conclusions about whether or not this line block is effective due to the lack of statistically significant differences between positive and negative controls. Indeed, the ability to assess pain in non-verbal species even with multiple assessment tools is called into question with the results of this study, necessitating a humble and compassionate approach to pain management in all non-verbal species.

The veterinary medical profession must work towards developing more sensitive and specific assessments of pain to evaluate the effectiveness of postulated analgesic interventions, while continuing to provide conscientious therapy knowing such strategies have not yet been developed. If an experienced observer cannot detect a patient with known pain from one that received adequate analgesia using four different techniques, it is relatively easy to miss a patient experiencing pain that cannot self-communicate. If one is inducing something that is likely to result in pain, aggressive pain management is warranted as a moral and ethical obligation.

Appendix 1

Site for line block/infiltration of local anesthetic or saline. Please see text for description of labels.

Additional Information and Declarations

Competing Interests

Author Contributions

Animal Ethics

Dr. McKune has no competing interests, but is employed by Mythos Veterinary LLC. Dr. Pascoe and Dr. Lascelles have no competing interests. Philip H. Kass is also an Academic Editor for PeerJ.

Carolyn M. McKune conceived and designed the experiments, performed the experiments, analyzed the data, contributed reagents/materials/analysis tools, wrote the paper, prepared figures and/or tables, reviewed drafts of the paper.

Peter J. Pascoe and B. Duncan X. Lascelles conceived and designed the experiments, contributed reagents/materials/analysis tools, reviewed drafts of the paper, and performed manuscript review.

Philip H. Kass conceived and designed the experiments, analyzed the data, and reviewed drafts of the paper.

The following information was supplied relating to ethical approvals (i.e., approving body and any reference numbers):

All protocols were approved by the University of California, Davis, Institutional Animal Care and Use Committee (IACUC): approval number 12597, as well as by administrative study reviewers at the Sacramento Society for Prevention of Cruelty to Animals (SSPCA).

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
