# Peer review of "The challenge of evaluating pain and a pre-incisional local anesthetic block"

_PeerJ, doi:10.7717/peerj.341_

## Round 0.1 · original submission · Major Revisions

Dear Dr McKune and colleagues,

Thank you for the submission of your manuscript to PeerJ.

I have carefully read your manuscript, and two reviewers with expertise in the subject area have also rendered opinions. In general, you have written a good manuscript investigating an important subject. However, there are several flaws in your paper, some of which are modifiable, and some which are not.

I would be grateful if you would examine the enclosed reviewers' comments, make the requested modifications, and submit a revision. Please ensure that you respond in detail to each suggestion, even those you decide not to incorporate it in a revision.

In addition to the Reviewers’ points, it is also important to address the following elements of your manuscript:

- L16: patients are not “painful”, but rather they are patients “with pain”
- L19: change “principle” to “principal”
- L31: as noted by the reviewer, re-read and fix grammar
- L36: change “improving” to “improve” and change “decreasing” to “decrease”
- L45: please define the first usage of “OVHX”. I realize this was defined in the Abstract, but it should be defined in the manuscript as well
- Lines 49-54 should be moved into your Methods section
- L167: inadequate information is provided to reproduce your sample size calculation. In addition to the detectable contrast, alpha, and beta, a SD is required
- L167: no discussion is made of performing a sample size calculation meant for two groups (with one between-group comparison of the primary outcome) and then applying the sample size calculation to three groups (with three potential between-group comparisons). The correct way of performing this sample size calculation would have been to modify your alpha depending on the number of planned comparisons of the primary outcome. This should be mentioned as a limitation, as it further contributes to the lack of power in this study.
- L171: P values should not be calculated on baseline data in a RCT since, by definition, even if a "significant" difference is found, it must be due to chance, which is inconsistent with the theory behind hypothesis tests. See Assmann SF, Pocock SJ, Enos LE, Kasten LE. Subgroup analysis and other (mis)uses of baseline data in clinical trials. Lancet 2000;355:1064-69. This applies to several variables you discuss, such as age, weight, etc.
- Why did you not present the baseline data (descriptive) in each of the groups in a Table?
- L183: which statistical test was used to compare these categorical data? Please include in your Methods section.
- Results: as mentioned by the reviewer, please ensure it is clear how many dogs were enrolled in each group
- L190-198 and L199-204 are essentially repeated paragraphs with slight differences. Please fix.
- L203: throughout the manuscript, when you state “…significant effect of time…” please state the direction of effect in the text
- L223: retrospective power analyses are not recommended. Please exclude this from your manuscript and discuss in a more general sense the lack of power in this study. Please see http://www.vims.edu/people/hoenig_jm/pubs/hoenig2.pdf for more information as to why retrospective power analyses are not helpful.
- Figure 1: It would be helpful to add horizontal lines at 1, 3, 5, etc. since these scores are all
below one. It would make it easier to compare scores.
- Figure 4 and 5: It would be helpful to the reader to explain the meaning of a lower (or higher) ratio. My understanding is that a lower ratio means that less force was applied to create pain, meaning that lower ratios are associated with tenderness at the application site of the algometer. Please clarify in the legend for these graphs.

Please note I can make no guarantee of acceptance after revision. Your revision will be peer-reviewed once again before a decision on publication is made. Thank you again for your submission to PeerJ.

Philip M Jones, MD MSc (Clinical Trials) FRCPC
* * *
Reviewer 1 ·

Basic reporting

No comments

Experimental design

Limitation of selected surgical model and sample size - see comments to authors, below.

Validity of the findings

Further investigation is required to understand the impact of surgeon ability/ experience on post-operative pain scores.

Additional comments

“The challenge of evaluating pain and a pre-incisional local anesthetic block”

The authors describe a prospective, randomized, blinded clinical trial investigating a range of pain scales for the assessment of pain in an ovariohysterectomy surgical model. Three treatment groups were compared, including proposed positive and negative control groups.

Major points:
The most striking finding of this study, as clearly identified by the authors, is the absence of significant difference between treatment groups. Two possible explanations for this include lack of statistical power and robustness of the surgical model.
1. The pain scores employed were able to differentiate changes over time, but not differences between treatment groups. The authors performed a retrospective power analysis indicating that sample size was sufficient to identify statistical differences. While the statistical analyses appear to have been performed appropriately it is unclear from the authors’ description to which pain scale the retrospective power analysis applied. The prospective power analysis indicated sufficient power to identify a difference of 2.6 with the GCPS with a sample size of 19 animals. The data (Fig 2) indicate mean differences smaller than this, suggesting that there was insufficient power to detect the observed differences.
2. The authors raise the issue of surgeon factors as a contributing factor the absence of significant differences between treatment groups. I would suggest that this was an important factor in light of the use of positive and negative control groups. Many studies on which evidence in favor of multimodal analgesia are based were performed in academic hospitals, where students performed at least part of the surgical procedure. Similarly, the GCPS was validated using a very different surgical population in a very different setting. Together, it is likely that the effect of surgeon ability has a greater role to play than previously recognized.
3. Though the study appears to have been well performed, I feel that the surgical model was minimally traumatic, limiting the ability of the pain scales and/ or sample size to identify significant differences. I would expect that if this study had been performed using student/ less experienced surgeons, the outcome would have been very different. The ability of the pain scales to identify changes over time indicate that they were adequately sensitive (data not shown so difficult to know if statistical significance was observed at all time points), further indicating a limitation of sample size and surgical model.
4. The authors raise an important issue in the discussion (lines 251+) - the generalizability of currently available pain scales to different surgical populations. This describes the concept of construct validation; ideally, though practically limiting, pain scales should be assessed in a range of patient populations at different institutions with different observers. To this reviewer’s knowledge, this has not been assessed for any of the pain scales employed.

Minor points:
1. Line 31 - check grammar
2. Line 189 - not clear how data were handled for dogs receiving rescue analgesia that were included in statistical analysis; were highest recorded pain scores (at time of rescue analgesia) carried forward or were animals assessed at each time period?

Reviewer 2 ·

Basic reporting

How many dogs were in each group? At what point were "initial power calculations" made?

Experimental design

No comments.

Validity of the findings

I agree with your assessment about why you may not have found a difference among the groups, and believe this requires more expansion.

With respect to surgeon experience, how experienced were these surgeons and what were the operative times? I can readily imagine an experienced shelter vet performing the operation through a very small incision would indeed induce minimal pain in the patient. In other studies using a negative control, what was the level of surgeon experience, and how do those pain scores compare to what you observed?

While I understand the strict adherence to the principle of rescue analgesia in the event of any patent having a maximum subscore on any of the Glasgow domains, this strategy led to a decrease in your effective sample size and may have confounded the results. After rescue medication, the rescued SS dogs effectively joined the LBM group, assuming the block did not provide appreciable analgesia, which has previously been demonstrated (Tsai et al. 2013). Is there precedent for treating patients who had a single domain of the Glasgow score at the maximum value? This is the first I have seen of that strategy, and it probably biased your results.

Additional comments

No comments.

---

## Round 0.2 · Minor Revisions

Dear Dr McKune and colleagues,

Thank you for the submission of your revised manuscript to PeerJ.

You have submitted an excellent revision which has adequately addressed each of the Reviewers’ points. Therefore, your manuscript is provisionally accepted to PeerJ. All that is now necessary is another minor revision to address the small points below.

I would be grateful if you would examine the enclosed reviewers' comments, make the requested modifications, and submit a minor revision. Please ensure that you respond in detail to each suggestion, even those you decide not to incorporate it in a revision.

In addition to the Reviewers’ points, it is also important to address the following elements of your manuscript:

- the Abstract on page 1 of the proofed PDF does not match the Abstract found on page 3 of the PDF. Please ensure they match when you re-submit
- there are still several areas in the manuscript (lines 104, 229, 350) where you refer to patients as “painful”. Please change to something like “with pain” or “experiencing pain”
- line 188: because you have specified an alpha and a beta later on, the text “with significance set at 0.05 and power set at 0.8” is now redundant. Please delete this (you can finish off the previous sentence with a period).
- line 233: change “full mu opioid” to “full mu opioid receptor agonist”
- Figures: add, in the caption, what the error bars are referring to (i.e. SD, SEM, 95% CI)

Contingent upon a satisfactory minor revision, your manuscript will be accepted for publication in PeerJ. Thank you again for your re-submission to PeerJ.

Philip M Jones, MD MSc (Clinical Trials) FRCPC
* * *
·

Basic reporting

No comments

Experimental design

No comments

Validity of the findings

No comments

Additional comments

Thank you for addressing my comments on the previous version of this manuscript. In future responses to reviewers it is helpful to include line numbers where changes have been made in response to specific comments.

Reviewer 2 ·

Basic reporting

Line 189 - Was this SD also provided from reference #13? If so, I would recommend making that clear. If not, the source of the variance should be noted.

I would suggest including the units in Table 1.

Experimental design

No comments.

Validity of the findings

Line 287-288 - Have other studies of OVH pain not used similar systems? Many other studies have used the OVH model and 'found' pain using these and similar systems.

---

## Round 0.3 · accepted · Accept

Thank you for submitting your revision. For some reason, the Abstract on the first page of the PDF still doesn't match what is in your Word document, but I will ask the PeerJ staff to copy and paste from your Word document into the Abstract field.